# Biosimilars in Oncology: Latest Trends and Regulatory Status

**DOI:** 10.3390/pharmaceutics14122721

**Published:** 2022-12-05

**Authors:** Deeksha Joshi, Rubiya Khursheed, Saurabh Gupta, Diksha Wadhwa, Thakur Gurjeet Singh, Sumit Sharma, Sejal Porwal, Swati Gauniyal, Sukriti Vishwas, Sanjay Goyal, Gaurav Gupta, Rajaraman D. Eri, Kylie A. Williams, Kamal Dua, Sachin Kumar Singh

**Affiliations:** 1Chitkara College of Pharmacy, Chitkara University, Rajpura 140401, India; 2School of Pharmaceutical Sciences, Lovely Professional University, Phagwara 144411, India; 3Delhi Pharmaceutical Sciences and Research University, New Delhi 110017, India; 4Department of Pharmaceutical Sciences, Amity University Lucknow, Lucknow 226028, India; 5Department of Pharmacology, KLE College of Pharmacy, Hubballi 580031, India; 6Department of Internal Medicine, Government Medical College, Patiala 147001, India; 7School of Pharmacy, Suresh Gyan Vihar University, Mahal Road, Jagatpura 333031, India; 8Department of Pharmacology, Saveetha Dental College, Saveetha Institute of Medical and Technical Sciences, Saveetha University, Chennai 602117, India; 9Uttaranchal Institute of Pharmaceutical Sciences, Uttaranchal University, Dehradun 248007, India; 10School of Science, STEM College, RMIT University, Melbourne, VIC 3001, Australia; 11Discipline of Pharmacy, Graduate School of Health, University of Technology Sydney, Ultimo, NSW 2007, Australia; 12Faculty of Health, Australian Research Centre in Complementary and Integrative Medicine, University of Technology Sydney, Ultimo, NSW 2007, Australia

**Keywords:** oncology, biologics, biosimilars, regulatory framework, traceability

## Abstract

Biologic-based medicines are used to treat a variety of diseases and account for around one-quarter of the worldwide pharmaceutical market. The use of biologic medications among cancer patients has resulted in substantial advancements in cancer treatment and supportive care. Biosimilar medications (or biosimilars) are very similar to the reference biologic drugs, although they are not identical. As patent protection for some of the most extensively used biologics begins to expire, biosimilars have the potential to enhance access and provide lower-cost options for cancer treatment. Initially, regulatory guidelines were set up in Europe in 2003, and the first biosimilar was approved in 2006 in Europe. Many countries, including the United States of America (USA), Canada, and Japan, have adopted Europe’s worldwide regulatory framework. The use of numerous biosimilars in the treatment and supportive care of cancer has been approved and, indeed, the count is set to climb in the future around the world. However, there are many challenges associated with biosimilars, such as cost, immunogenicity, lack of awareness, extrapolation of indications, and interchangeability. The purpose of this review is to provide an insight into biosimilars, which include various options available for oncology, and the associated adverse events. We compare the regulatory guidelines for biosimilars across the world, and also present the latest trends and challenges in medical oncology both now and in the future, which will assist healthcare professionals, payers, and patients in making informed decisions, increasing the acceptance of biosimilars in clinical practice, increasing accessibility, and speeding up the health and economic benefits associated with biosimilars.

## 1. Introduction

Biological therapeutic agents—or biologics—comprise a wide range of substances that are manufactured by cells or living organisms via various biological processes, such as controlled gene expression, antibody technologies, or recombinant DNA technologies [1]. In recent decades, the management of various acute and chronic diseases—such as hormonal deficiencies, a range of autoimmune and inflammatory diseases, and solid tumors and hematological malignancies—has been reshaped by biologics [2]. Moreover, in the pharmaceutical market, half of the products available for the management and treatment of malignancies are biologics [2]. However, developing biological drugs is expensive and time-consuming; therefore, the whole biopharmaceuticals industry is shifting its focus to developing “Biosimilars” [3,4]. Biosimilars are biopharmaceutical products that highly resemble already existing reference products in terms of safety, potency, and purity [5]. Despite having similar amino acid sequences to their reference products, biosimilars may still have distinctive attributes such as their 3D structures, isoform profiles, glycosylation sites, and protein aggregation. Biosimilar products, unlike generic medicines, are not considered to be exact copies of the reference products; however, many of their parameters—such as their mechanism of action, dosage form, route of administration, therapeutic indication, and strength—must be identical to those of the reference product [6,7,8,9].

The approach to managing and treating cancer using biological agents was initiated in 1980. The first biopharmaceutical was approved by the Food and Drug Administration (FDA) in June 1986, named Interferon alfa-2b (INTRON ATM, Schering Corporation, Kenilworth, NJ, United States of America (USA)), which currently holds seven labels [10]. Subsequently, the European Medicines Agency (EMA) was the very first regulatory body to initiate the guidelines for biosimilars in October 2005 [11]. It has been noted that the introduction of biosimilars has led to a 44% increase in the number of patients who have access to cancer treatments and medications [12]. Filgrastim (Zarxio^®^) was the first biosimilar approved in the USA in 2015. Moreover, in 2015, the FDA introduced the guidelines associated with biosimilars, addressing scientific and quality aspects in showing biosimilarity to the original product. The objective was to provide guidance about the FDA’s approach to determining biosimilarity and to assist in obtaining license for the market. Furthermore, in 2016 and 2018, the FDA issued guidance documents intended to address issues such as clinical pharmacological evidence to substantiate biosimilarity and labeling guidelines for inclusion in an application within section 351(k) of the Public Health Service Act (42 U.S.C. 262(k)).

There are many bottlenecks that have prevented biosimilars from being completely accepted in the clinical practice to date. For instance, immunogenicity is the main concern of biosimilars, and these immune reactions tend to have side effects, with the majority of them impacting the product’s efficacy. Therefore, continuous monitoring of safety and efficacy is required in both the clinical trials and post-marketing phases [13]. Another challenge in the case of biosimilars is lack of awareness. Research has revealed that there is a need to create awareness among healthcare professionals so that they can understand the concept of biosimilars based on reliable scientific data—such as clinical trial data—in order to bring biosimilars into clinical practice [14]. In one study, Cook et al. reported that only 26% of oncologists and 21% of practitioners are aware of the idea of biosimilars [13]. Many regulatory bodies have set up standardized guidelines and approval procedures for biosimilars, but switching from the high-cost biologics to the low-cost biosimilars is still the main concern for patients as well as for healthcare professionals—mainly in terms of safety [15]. Biosimilars’ safety and efficacy in extrapolated indications has also been a point of contention for providers, with over 39% of rheumatologists disagreeing with the extrapolation of indications for biosimilars in a 2017 survey [16]. In the United States and Europe, interchangeability and pharmacy-level substitution are another source of concern, along with lack of awareness among providers; in 2016, 89.9% of surveyed clinicians opposed pharmacy-level substitution of biosimilars, and then in another 2016 survey 80% of participants were unaware that an interchangeability designation could result in automatic substitution [16].

This article’s main goal is to provide a complete summary of the biosimilars used in cancer in one place, covering the numerous biosimilar choices available to date, their respective regulatory requirements around the world, and the most common adverse events connected with them. Biosimilars’ primary advantages and disadvantages, in addition to their existing and future positions in medical oncology and the various associated challenges, are also addressed. This article will assist readers in understanding worldwide regulatory requirements and identifying regulatory loopholes, allowing authorities to alter their guidelines and streamline the clearance process while focusing on safety. We define recent patterns in the emergence of biosimilars in cancer, providing a full account of the different issues faced by pharmacovigilance experts, as well as recommendations for how to overcome them. Furthermore, this review aims to provide comprehensive knowledge about biosimilars for physicians, other healthcare professionals, payers, and patients to enable them to make informed decisions, as well as to boost biosimilars’ acceptance in clinical practice, increase their accessibility, and speed up the related health and economic benefits.

## 2. Healthcare Burden of Cancer

### 2.1. Incidence and Prevalence

In every country, cancer is a primary cause of death as well as a significant impediment to extending the average lifespan. By the end of 2020, an estimated 19.3 million new cancer cases (18.1 million excluding non-melanoma skin cancers) were diagnosed worldwide, with around 10.0 million cancer-related deaths (9.9 million excluding non-melanoma skin cancer). Breast cancer (BC) has surpassed lung cancer (11.4%) as the most frequently diagnosed malignancy among women, with an estimated 2.3 million new cases (11.7%), followed by colorectal (10.0%), prostate (7.3%), and stomach (5.6%) cancers. Lung cancer remained the most common cause of death from cancer, with a projected 1.8 million fatalities (18%), followed by colorectal (9.4%), hepatic (8.3%), gastric (7.7%), and women’s breast (6.9%) cancers. Overall, the incidence in both sexes was 2–3-fold higher in transitioned vs. transitional countries, whereas death was 2-fold higher for males and marginally higher for females. In contrast, cervical cancer and female BC death rates were substantially higher in transitioning countries than in transitioned regions (12.4 vs. 5.2 per 100,000 and 15.0 vs. 12.8 per 100,000, respectively). Due to demographic changes, the global burden of new cancer cases is predicted to reach 28.4 million cases in 2040, up 47% from 2020, with a greater increase in transitional countries (64–95%) versus transitioned countries (32–56%). However, increased risk factors connected with global competition and an expanding economy may worsen this impact. Efforts to develop a sustainable infrastructure for preventative measures against the spread of cancer, to reduce the financial burden of chemotherapy (CHM), and to provide cancer care in transitioning countries are critical for global cancer control [17].

The most prevalent malignancies detected in males and females in 2021 are depicted in Figure 1. Men’s malignancies of the prostate, lungs, and bronchus, as well as colorectal cancers (CRCs), accounted for 46% of all new occurrences, with prostate cancer accounting for only 26%. BC, lung cancer, and CRC accounted for half of all new cancer cases in women, with BC accounting for 30% of all malignancies in women [18].

### 2.2. Risk Factors

Cancer is more likely to occur as people get older; in the United States, 80% of all malignancies are diagnosed in people aged 55 years and older. Certain habits and other changeable variables—including obesity; smoking; alcohol consumption; viral infections such as human papillomavirus (HPV); specific chemicals such as aflatoxins, arsenic, asbestos, etc.; radiation exposure, including ultraviolet radiation from the Sun; and eating an unhealthy diet—are all contributing factors to the pathogenesis of cancer. An estimated 41 out of every 1000 males and 39 out of every 1000 females in the United States will be diagnosed with cancer at some point during their lives [19]. However, this estimation is based on the incidence of cancer in the general public and may vary due to differences in exposure to risk factors (e.g., smoking), family background, and/or genetic predisposition. Some human infections are also risk factors for cancer, which is a major problem in middle- and low-income nations. Carcinogenic infections such as Helicobacter pylori, Epstein–Barr virus, human papillomavirus (HPV), and the hepatitis B and C viruses were responsible for almost 13% of malignancies diagnosed globally in 2018 [19]. Some kinds of HPV, as well as the hepatitis B and hepatitis C viruses, can increase the risk of cervical cancer and liver cancer. HIV infection significantly increases the risk of malignancies such as cervical cancer [19].

### 2.3. Treatment

Since the pathogenesis of each cancer is different, every type of cancer will require a different treatment plan. Moreover, an accurate cancer diagnosis is critical for suitable and effective treatment. Radiotherapy, CHM, and/or surgery are commonly used in the treatment of cancer. The first step in treatment is to figure out what we want to get out of it. The main objective of any cancer treatment is to either cure cancer or significantly extend the patient’s life. Another important purpose is to maximize patients’ quality of life. Support for the patient’s emotional, physical, and spiritual wellbeing, as well as palliative care in the latter stages of cancer, can help achieve this. Colorectal cancer, cervical cancer, BC, and oral cancer, for example, all have high functional recovery when detected early and treated properly. Even though cancerous cells are found in other parts of the body, several forms of cancer—including testicular seminoma and other forms of leukemia and lymphoma in adolescents—have good cure rates if adequate treatment is administered [19].

## 3. Pharmacoeconomics of Biosimilars

The rising cost of cancer treatment is putting strain on national healthcare budgets, posing a threat to overall healthcare funding. The burden due to the increased cost of medicines and hospital charges continues to climb. As the prevalence of various cancers rises, cancer medications may be started earlier due to advances in early cancer identification, and treatment may be extended due to increased patient survival, further increasing the cost burden. Oncology spending increased by 15–22% in health plans from 2018 to 2019 [20]. Currently, many cancer medications are in the pipeline for clinical studies, and it has been predicted that the oncology category will grow by 105% by 2024 (see Figure 1). Some chemotherapeutics and targeted therapies have continued to surge in price. For example, the cumulative cost rise was 29% for bevacizumab, 78% for trastuzumab, and 85% for rituximab from 2005 to 2017 (see Figure 2 and Figure 3). Biosimilars for all of these biologics represent a significant possibility to reduce healthcare costs and increase accessibility to even more patients with advanced cancer [21]. Biosimilars could save the US health system USD 54 billion within the next 10 years, according to a recent RAND (Research and Development) group report, with possible savings ranging between USD 25 billion and 150 billion [20].

The patients are not able to pay the costs of treatment as healthcare expenses rise. Oncology biologics may cost more than USD 10,000 per month, which is nearly twice the US monthly average earnings [22]. Due to their complicated physicochemical properties, biologics are more expensive to produce than standard CHM medicines, and this has influenced the expense of cancer treatment. Payers may explore promoting the use of low-cost therapeutic regimens that yield the same degree of recommendation by compendia. Biosimilars offer the potential to help cut healthcare costs while maintaining high-quality cancer therapy, since there are no clinically significant differences between them and the reference products [22].

Until the BPCIA was implemented in 2010, most biosimilars were unable to reach the market due to patent litigation. Biosimilars made up just under 2% of the entire biologics market in the United States in 2018. Both the number of FDA approvals and the use of biosimilars increased dramatically in 2019. Bevacizumab, trastuzumab, and rituximab were the top three biosimilars launched in 2019, with bevacizumab leading at 42%, followed by trastuzumab at 38% and rituximab at 20%, and their utilization is further rising [22]. In comparison, filgrastim—the very first biosimilar—had a market share of only 25% within the first year of approval. This jumped to 80% of the total in 2019, indicating that biosimilars are becoming more widely available. The potential cost impact of biosimilars is frequently contrasted to European procedures, as the European Medicines Agency has now authorized the greatest number of biosimilars in the world since introducing biosimilar regulations in the early 2000s. The introduction of biosimilars has enhanced preventive care and treatment, potentially leading to improved clinical results. Adoption in medical practice is projected to rise as additional biosimilars become accessible following regulatory approval [22].

Biosimilars have been shown to save money in absolute terms, with higher savings from more recent launches when originators were more expensive, as depicted in Figure 4 below.

For example, when a new product is compared to an existing standard treatment, the incremental cost-effectiveness ratio is defined as the ratio of the change in cost of a therapeutic intervention. The effect of switching interventions is referred to as incremental. The number of quality-adjusted life years (QALYs) gained by the intervention is frequently used to quantify the change in effects [23]. This includes both disparities in health while alive and years gained or lost due to a differing time of death. In a recent publication, researchers looked at the threshold incremental cost-effectiveness ratios of biosimilars in a number of nations, according to which the quality-adjusted life year threshold values for Canada, England, the Netherlands, New Zealand, and the Unites States ranged from EUR 1400 to 80,000, while Australia had a threshold value of EUR 24,700 to 44,700 per life year [23].

The economic analysis of biosimilars used in the treatment of cancer is still in its early stages around the world. However, the affordability and the cost-effectiveness of biosimilars used in oncology is actively supported by the minimal evidence available from developed nations, despite the fact that uptake and price discount rates are major factors [24].

## 4. Current Impact of Biosimilars in Cancer Care

The use of drugs to counteract the side effects of cancer treatment is known as supportive cancer therapy. The first approved biosimilars in the European Union (EU) were supportive care therapies—filgrastim and epoetin alfa—in 2007. Many biosimilar development programs, such as those for rituximab, trastuzumab, and bevacizumab, have been launched since then. Filgrastim and epoetin are two biologic medications with considerable cancer-fighting benefits that have had their patents expire. Since 2007, the European Medicines Agency (EMA) has approved biosimilar variants of both in Europe. Using a general approach, this could expand accessibility to improve supportive patient care while also saving money, allowing the existing funding to be reassigned to new treatment areas [25]. A number of biosimilars for filgrastim have already been approved for all of the reference product’s indications. The authorization was based on information that included accurate comparisons to the reference standard, utilizing analytical methods to show its structure, in vitro parameters, PK and PD parameters, and mode of action similarities. In addition, investigations of their effectiveness and safety in cancer patients were carried out [26].

The most important goal of health policy is to maintain the population’s health. Pharmaceutical therapy is the most prevalent medical treatment in modern healthcare systems. The efficacy of public health initiatives in improving the overall health of the population is strongly dependent on the performance of policies aimed at decreasing medicine prices and enhancing patient access [27]. The WHO clearly indicates that appropriate prescription requires finding and using affordable drugs.

Biosimilars, like generic pharmaceuticals, may require an active strategy by payers to promote their use in order to receive the optimal returns for a health system. These could be in the form of payer directives, patient copayments, or physician gain-shares. Poland, Finland, Denmark, and the Netherlands have already developed national plans for moving to biosimilars. This endeavor is critical to consider now, because 9 of the world’s top 10 best-selling medications are set to lose patent protection in the next five years, with combined sales of more than USD 50 billion in 2011 [28]. The cost of expensive medications used to treat conditions such as cancer and rheumatoid arthritis may decrease if biosimilar treatments are substituted for biologics; savings are predicted to total USD 38.4 billion, or 5.9% of the total projected US expenditure on biologics from 2021 to 2025, according to a new RAND Corporation study [29]. Therapeutic substitution, also known as drug swapping or therapeutic exchange, offers a possible gain in addition to shifting between standard and biosimilar pharmaceuticals. This is the practice of substituting chemically different medications for a patient’s prescription drugs that are believed to have had the same clinical impact. Prescribers should explore therapeutic replacement under departmental norms given that short-acting filgrastim and epoetin are accessible as less-expensive biosimilars [28].

Biosimilars will have a big impact in oncology on a global scale. The WHO’s model lists of essential medicines for children and adults represent medicines that are both cost-effective and vital to public health in all countries. The approval of biosimilars for filgrastim in places such as the EU, the United States, and Japan, as well as the patent expiration and anticipated impending arrival of biosimilars for trastuzumab and rituximab, has allowed all three medications to be added to the most recent WHO essential pharmaceuticals list. The WHO states that these pharmaceuticals should be available at all times in sufficient quantities and in acceptable dose forms, at a cost that the community can afford, which is critical to their inclusion in the essential drugs list [26].

## 5. Current Biosimilars Used in Oncology Globally

Regulatory agencies around the world—including the European Medicines Agency (EMA), the Food and Drug Administration (FDA), the Therapeutic Goods Administration (TGA) in Australia, and Health Canada in Canada—have approved a few biosimilars in the field of oncology [30]. Biosimilars can be used for various indications and in many therapeutic areas; for example, the biosimilars filgrastim and epoetin are utilized in the treatment of CHM-induced NTP and for the treatment of ANE induced by CHM in cancer patients [30,31]. The small alteration in the original biological product can lead to the loss of efficacy and facilitates the occurrence of adverse events that induce immunogenicity, and these are the main concerns that oncologists are facing in switching from original biological products to biosimilars. There are many ways by which immunogenicity can occur, such as distinctive routes of administration, patient characteristics, and storage setup [32]. Moreover, a limited number of studies have evaluated the utilization of biosimilars in the field of oncology due to the arduous nature of biosimilars [33]. Despite many studies having been conducted on biosimilars of rituximab for the management and treatment of rheumatoid arthritis (RA), there are many uncertainties as to whether or not rituximab can be extrapolated in oncology [33]. Risks associated with biosimilars can be detected by preclinical studies in which original biological products are compared with biosimilars, along with conducting post-marketing surveillance studies in order to evaluate their benefit–risk profiles in clinical practice [15,34]. Furthermore, data obtained by pharmacovigilance departments are essential to monitor the safety and efficacy of any biosimilar in long-term clinical practice, and such information related to adverse events—including medication errors and safety concerns—must be reported by healthcare professionals [35].

The majority of biosimilars have been marketed in Europe for various indications since 2006 for the management and treatment of cancer. The EMA is the decentralized regulatory authority in the European Union, whose framework for biosimilars is also followed by other countries. The first biosimilar approved under the EMA was Abseamed, produced by the marketing authorization holder Medice Arzneimittel Putter GmbH & Co. Kg, as depicted in Table 1. The FDA is another regulatory authority with a framework for biosimilars, which was approved in the USA in 2009 when the Biologics Price Competition and Innovation Act was passed [35]. Currently, biosimilars have proven to be safe and efficacious in the last 10 years, and recently many biosimilars have been approved globally for the management and treatment of cancer, as demonstrated in Table 1. In the near future, there are many biological products for which biosimilars are expected to be marketed—not only for the management of cancer, but also for other diseases such as Crohn’s disease, colitis, psoriasis, RA, and other autoimmune diseases.

## 6. Biosimilars in Oncology Approved by Various Regulatory Authorities

BC, stomach cancer, CRC, and other malignancies can all be treated with biosimilars. They can also be used to treat cancer-related side effects such as decreased white blood cell counts, which raise the risk of infection. Various cancer-related biosimilars that have been approved globally are listed below.

### 6.1. Epoetin

There are various indications associated with epoetin and its analogues, as mentioned in Figure 5, Figure 6, Figure 7 and Figure 8. There is a hormone called erythropoietin, which is secreted by the kidneys in order to prompt the synthesis of bone marrow’s red blood cells, and in the event of the inability of the body to create sufficient EPO to generate red blood cells, manmade versions of natural erythropoietin known as erythropoiesis-stimulating agents (ESAs) can be used. These ESAs mitigate the need for constant drug transfusions in CHM [80,81]. Many studies have shown that ESAs are generally well accepted by patients [82]. In Europe, five epoetin alfa biosimilars have been approved, along with three epoetin lambda biosimilars approved in Australia and one biosimilar of epoetin alfa-epbx marketed in the USA. EPO-alfa is used for treating the ANE induced by CHM received in patients suffering from malignant lymphoma, solid tumors, and multiple myeloma [30]. Unlike epoetin lambda, EPO-alfa is higher in efficacy due to its conformational structure [83].

### 6.2. Filgrastim and Pegfilgrastim

Filgrastim is a stimulating factor for recombinant granulocyte colonies that stimulates bone marrow for the synthesis of neutrophils. The main function of filgrastim is to facilitate the counts of peripheral blood neutrophils within a day, which further facilitate the counts of monocytes [84,85]. Filgrastim is utilized to decrease the occurrence and time period of FN in patients who are suffering from non-myeloid malignancies and receiving treatment in the form of myelosuppressive anticancer drugs [31]. The first biosimilar of filgrastim was approved in 2008 in Europe; since then, seven biosimilars have been approved in Europe and three biosimilars have been approved in Australia by the TGA. Each biosimilar that was approved in USA and Canada in 2015 is mentioned in Figure 6 and Figure 8, respectively. Pegfilgrastim and its analogues have an additional unit of polyethylene glycol, which results in increasing the size of the molecule and, hence, increases the half-life of the biopharmaceuticals. Six pegfilgrastim biosimilars were approved in Europe in 2018; more recently, in 2020, there were another three biosimilars marketed, as well as one biosimilar to pegfilgrastim that was approved in Canada in 2020. These are summarized in Figure 5, Figure 6, Figure 7 and Figure 8.

### 6.3. Rituximab

Rituximab was the first biosimilar monoclonal antibody, primarily authorized in Europe for oncological treatment [33]. It is a genetically engineered chimeric monoclonal antibody that binds selectively to the non-glycosylated phosphoprotein transmembrane antigen CD20, which is present on pre-B and mature B cells. Over 95% of all B-cell non-Hodgkin lymphomas express this antigen. When an antibody binds to this antigen, it does not assimilate and does not leave the cell surface; hence, it does not compete with other antigens for antibody binding as a free antigen in the plasma. The Fc domain of rituximab can mobilize immunological effector functions to cause B-cell lysis, and the Fab domain of rituximab binds to the CD20 antigen on B cells; therefore, it has been utilized for the treatment of n-HL and chronic lymphocytic leukemia [85]. Recently, in 2017, rituximab was initially approved in Europe and Australia, and five rituximab biosimilars have been approved in Europe to date, along with two biosimilars that have been approved in Australia. Moreover, in 2020, two biosimilars were approved by Health Canada. Additionally, biosimilars for two analogues of rituximab (rituximab abbs and rituximab pvvr)—known as Truxima and Ruxience—were approved in the USA in 2018 and 2019, respectively [86,87].

### 6.4. Trastuzumab

In 2017, the first biosimilar of trastuzumab was approved in Europe, known as Ontruzant; subsequently, the USA approved a biosimilar named Ogivri in late 2017. The indications of trastuzumab are distinctive; for example, Ontruzant can be utilized as a monotherapy or can be combined with other drugs in CHM for the management and treatment of human epidermal growth factor receptor 2 (HER2)-positive metastatic cancer, whilst Ogivri is indicated for the management and treatment of EBC, MGC, and MBC [87]. Ogivri must be used in combination both with cisplatin and with 5-fluorouracil or capecitabine. The mechanism of action of trastuzumab involves binding with HER2, which tends to inhibit the ligand-dependent HER2 signaling. This process prevents the activation of HER2. Furthermore, proliferation of tumor cells, which can further activate HER2, can be inhibited by trastuzumab; hence, it is considered to be a formidable mediator of antibody-dependent cell-mediated cytotoxicity [88].

### 6.5. Bevacizumab

Bevacizumab was originally approved by the FDA in 2004 and utilized as a first-line treatment for metastatic CRC in combination with fluorouracil-based CHM; however, in 2017 and 2018, the FDA and EMA approved their first biosimilars of bevacizumab, named Mvasi and Avastin, respectively [89], as shown in Table 1. Bevacizumab is a recombinant humanized monoclonal antibody that further binds and impedes vascular endothelial growth factor (VEGF) [90]. This process blocks the angiogenesis of newer tumors and the growth of tumor cells. There are various indications for which Mvasi can be utilized when used in combination therapies, including advanced or metastatic renal cell carcinoma; recurrent, persistent, or metastatic carcinoma of the uterine cervix; metastatic colorectal carcinoma; recurrent glioblastoma; and unresectable, recurrent, locally advanced, metastatic non-squamous non-small-cell lung cancer [91]. The detailed description of indications associated with biosimilars in different countries is given in Figure 5, Figure 6, Figure 7 and Figure 8.

### 6.6. Recombinant Human Interferon-α-2α

A biosimilar of interferon named Roferon-A was originally approved in the USA for the management and treatment of hairy-cell leukemia, follicular n-HL, cutaneous T-cell lymphoma, chronic-phase Philadelphia-chromosome-positive myelogenous leukemia, advanced renal cell carcinoma, and stage II malignant melanoma. Interferons are a group of proteins that induce signaling processes and can be synthesized in the body in order to fight against infections such as viruses [88]. However, a biosimilar called Alpheon was proposed by BioPartners GmbH in 2006, but the EMA refused to market it. This was due to the vast differences in the results during the comparative studies of the reference biological product and Alpheon. Furthermore, the safety data were inadequate and the risk-to-benefit ratio of Alpheon was higher than the standard requirements [92].

**Figure 5 pharmaceutics-14-02721-f005:**
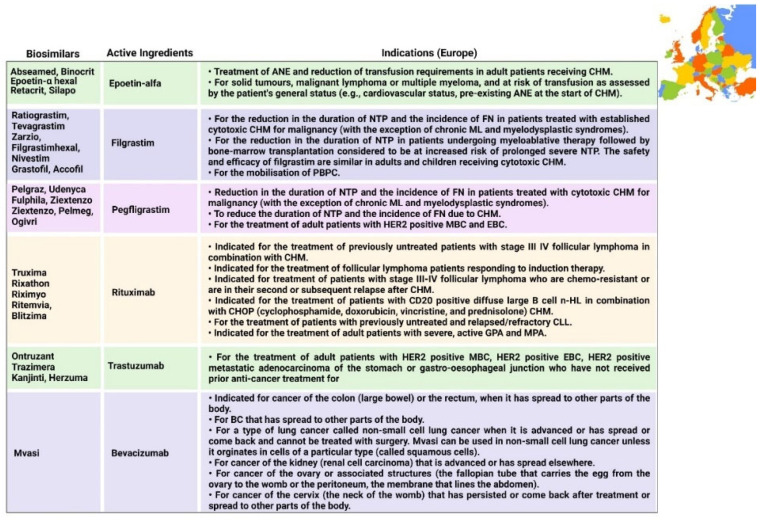
Indications approved in Europe for biosimilars [93,94,95,96,97,98,99,100,101,102,103,104,105,106,107,108,109,110,111,112,113,114,115,116,117,118]. ANE = anemia; CHM = chemotherapy; CHOP = cyclophosphamide, doxorubicin hydrochloride (hydroxydaunorubicin), vincristine sulfate (Oncovin), and prednisone; CLL = chronic lymphocytic leukemia; EBC = early breast cancer; FN = febrile neutropenia; GPA = granulomatosis with polyangiitis; HER2 = human epidermal growth factor receptor 2; MBC = metastatic breast cancer; ML = myeloid leukemia; MPA = microscopic polyangiitis; PBPC = peripheral blood progenitor cell; n-HL = non-Hodgkin lymphoma; NTP = neutropenia.

**Figure 6 pharmaceutics-14-02721-f006:**
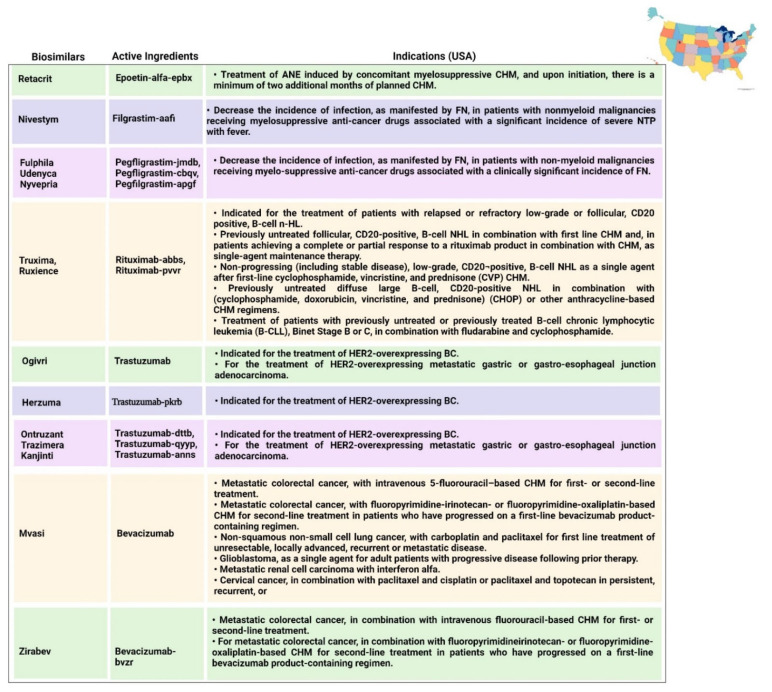
Indications approved in the USA for biosimilars [40,42,52,58,59,60,66,67,69,73,74,75,76,77,79,119,120,121,122,123,124,125]. ANE = anemia; B-CLL = B-cell lymphocytic leukemia; CHM = chemotherapy; NTP = neutropenia; FN = febrile neutropenia; ML = myeloid leukemia; PBPC = peripheral blood progenitor cell; HER2 = human epidermal growth factor receptor 2; MBC = metastatic breast cancer; EBC = early breast cancer; n-HL = non-Hodgkin lymphoma; CLL = chronic lymphocytic leukemia; GPA = granulomatosis with polyangiitis; MPA = microscopic polyangiitis; CHOP = cyclophosphamide, doxorubicin hydrochloride (hydroxydaunorubicin), vincristine sulfate (Oncovin), and prednisone.

**Figure 7 pharmaceutics-14-02721-f007:**
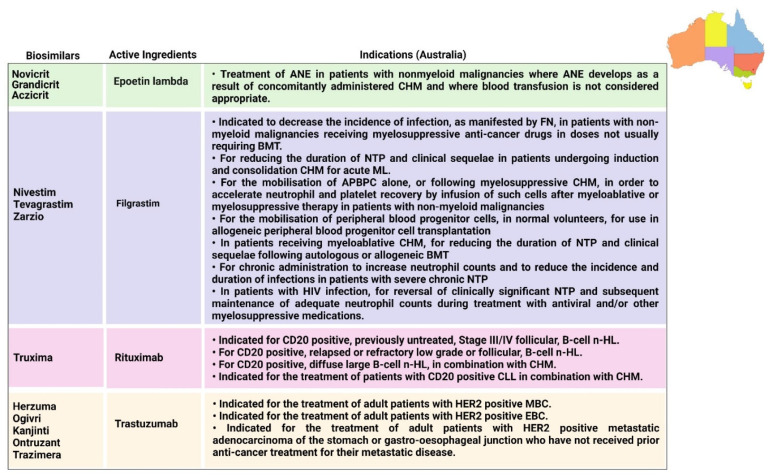
Indications approved in Australia for biosimilars [126,127,128,129,130,131]. ANE = anemia; BC = breast cancer; APBPC = autologous peripheral blood progenitor cell; B-CLL = B-cell chronic lymphocytic leukemia; BMT = bone marrow transplantation; CHM = chemotherapy; CLL = chronic lymphocytic leukemia; EBC = early breast cancer; FN = febrile neutropenia; HER2 = human epidermal growth factor receptor 2; ML = myeloid leukemia; MBC = metastatic breast cancer; n-HL = non-Hodgkin lymphoma; NTP = neutropenia.

**Figure 8 pharmaceutics-14-02721-f008:**
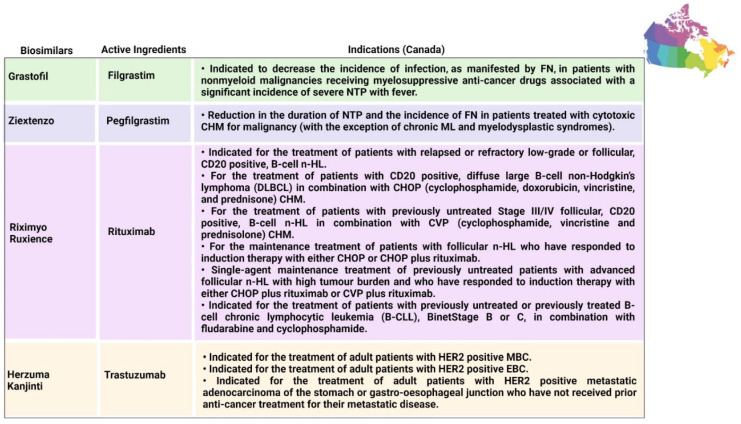
Indications approved in Canada for biosimilars [132,133,134,135,136,137]. B-CLL = B-cell chronic lymphocytic leukemia; CD20 = cluster of differentiation 20; CHM = chemotherapy; CHOP = cyclophosphamide, doxorubicin hydrochloride (hydroxydaunorubicin), vincristine sulfate (Oncovin), and prednisone; CVP = cyclophosphamide, vincristine, and prednisone; DLBCL = diffuse large B-cell non-Hodgkin lymphoma; EBC = early breast cancer; FN = febrile neutropenia; HER2 = human epidermal growth factor receptor 2; MBC = metastatic breast cancer; ML = myeloid leukemia; n-HL = non-Hodgkin lymphoma; NTP = neutropenia; PBPC = peripheral blood progenitor cell.

## 7. Regulatory Guidelines Associated with Biosimilars Globally

The rules and regulations of biosimilars are improving globally, particularly in the USA and Europe, in order to attain harmonization all over the world. There are a few main regulatory authorities, such as the EMA, FDA, and WHO, which have set up standards to determine the similarity between the reference products and biosimilars. Many regulatory agencies require comparative studies to evaluate the safety, quality, and efficacy of biosimilars in oncology. In comparison to the small-molecule drugs, biosimilars are more complex in nature, which makes them more subtle. The first regulatory authority to provide a framework for biosimilars was the EMA back in 2003, whereas the primary approval for such products was given in 2006. In 2009, the FDA standardized the specific guidelines for the approval of biosimilars [138]. Moreover, since 2008 and 2012, many other countries—such as Canada, Japan, India, South Korea, and Australia—have also produced their own guidelines, which are analogous to the EMA guidelines. Recently, China and Russia have been setting up standardized regulatory pathways; however, they have been approving biosimilars as new biologics thus far [139]. Furthermore, there are many developing countries that do not have regulatory pathways for biosimilars, so they allow extrapolation to indications that have been approved previously. The various existing regulatory procedures across the world are discussed below.

### 7.1. Europe

According to the EMA guidelines, there is a streamlined approval process available for biosimilars that is based on preclinical and clinical studies that compare their efficacy, safety, and immunogenicity with the reference biological products [140]. There must be a previously approved biological product under EMA regulations for at least 10 years, and after the patent expires the biosimilars can be approved. There are many classes of biosimilars in oncology, such as epoetins and filgrastims, and the regulatory guidelines are customized according to the product [141,142]. There are different data requirements for the approval process depending on the types of cases associated with the reference product [120]. The EMA regulations address toxicology, non-clinical pharmacology, pharmacodynamics, pharmacokinetics, manufacturing, and clinical considerations. There are certain conditions of biosimilars—such as their route of intervention, form, and strength—that must be identical to those of the reference products, and their comparative quality as well as non-clinical toxicological data must be analyzed.

Various studies are also required in the approval process, such as pharmacodynamic and pharmacokinetic data on the clinical efficacy of biosimilars, which are further evaluated by two or three groups of clinical potency and safety findings. Human studies are also required for the approval, involving the reference biological product, biosimilars, and a placebo. However, safety, immunogenicity, and efficacy can be determined by performing some specific comparative analyses. In the case of biopharmaceuticals (e.g., biosimilars and biological products), adverse events tend to occur after several years, so continuous surveillance is required, which is performed via post-marketing and risk management studies. In some cases, extrapolation of additional indications can be permitted, but this depends on the nature of the cases.

### 7.2. World Health Organization (WHO) Guidelines

The regulatory framework for generic medications is not adequate for the evaluation and licensing of similar biotherapeutic products (SBPs), as many nations’ regulatory agencies are well aware. The WHO’s Expert Committee is responsible for publishing the regulatory regulations “Guidelines on Evaluation of SBPs”, which standardize the biopharmaceuticals. It is mandatory for all biotherapeutics to follow this framework in order to assure the safety, quality, and efficacy of the approved biopharmaceuticals. Like the EMA, the WHO has also developed a regulatory framework for the approval of SBPs that evaluates comparative studies on the quality, non-clinical parameters, and clinical parameters with those of the original biopharmaceutical. Biosimilars must be comparable in terms of dosage and route of intervention, and they must be granted a license based on a complete registration dossier [143].

### 7.3. USA

The USA successfully set up standards via the Biologics Price Competition Act (BPCIA) for the approval of biosimilars in 2009 [121]. For the approval of any biosimilar, the reference product must be on the market at least for 12 years [95]. In certain cases, extrapolation to indications can be possible depending on the nature of the case. The FDA must evaluate some data based on comparative analysis by utilizing preclinical and clinical studies, as well as functional and physiochemical assays that can alter the synthesis process. From 2010 to 2014, the FDA produced four drafts. The main objective of these guidelines is to identify certain outlines in order to analyze the profiles of various biosimilars in terms of efficacy, safety, potency, and purity by assessing their function, structure, toxicity in animals, and pharmacodynamics and pharmacokinetics in human trials. The guidelines also reveal the requirements for the evaluation and analysis of biosimilars’ safety, efficacy, and immunogenicity in human trials. The overall risk–benefit assessment is entirely based on the evidence gathered from various studies that have been performed under these guidelines [144,145].

There are certain FDA guidelines that deal with comparative studies of biosimilars and their reference biological products that have been licensed outside USA. These studies include synthesis processes and protein complexity. However, it could be worth considering that certain elements of regulatory strategies might not be necessary. In August 2014, for biosimilars, two biologic licensing applications (BLAs) were submitted via the 351(k) biosimilars strategy [119]. Scientific data can be exchanged through EMA–FDA biosimilar collaborative meetings. This collaboration can be beneficial for the FDA, who can take all of the lessons learned by the EMA and utilize them for developing more standardized and accurate guidelines, just as the EMA made guidelines separately for filgrastim and epoetin [122].

### 7.4. Australia

In 2008, the EMA guidelines were also adopted by the regulatory authority of Australia—the Australian Therapeutic Goods Administration (TGA). During adoption, the EMA’s requirements for the extrapolation of indications were not taken verbatim. However, the requirements for comparative studies were taken verbatim from the EMA guidelines, and some data requirements were also adopted from the International Conference on Harmonization [123]. The reference biological product must be marketed in Australia in order to be approved as a biosimilar in Australia. Australia provides only 5 years of data protection for novel biologics [127].

### 7.5. Canada

Health Canada is a regulatory authority that standardized some guidelines for the approval of biosimilars in Canada in 2010. These guidelines were drafted in such a way that the approval process of biosimilars is entirely dependent on the similarity between the reference products and the biosimilars [123]. A biosimilar must follow a biologic medicine that is approved in Canada and to which a reference is made, according to the guidelines. In comparative studies, sponsors may utilize a non-Canadian-sourced version as a proxy for the Canadian medicine. The sponsor is responsible for demonstrating that the reference biologic medication chosen is appropriate for the submission. Among the factors to consider while using a non–Canadian reference biologic medicine is that it must have the same medicinal ingredient(s), dosage form, and method of administration as the one approved in Canada. The Drug Product Database of Health Canada has information on the Canadian version [123].

The non-Canadian reference biologic drug should be marketed in a country that has formally adopted the International Council for Harmonization (ICH) guidelines and has regulatory standards and principles for medicine evaluation, post-market surveillance activities, and comparison approaches that are comparable to those in Canada [123,132].

If a non-Canadian reference biologic medication is used in clinical trials in Canada, data must be submitted to meet chemistry and manufacturing (quality) requirements under the Food and Drug Regulations, section C.05.005. Herzuma (trastuzumab) by Celltrion Healthcare (intended for metastatic and gastric cancer), Kanjinti (trastuzumab) by Amgen (intended for BC), and Mvasi (bevacizumab) by Amgen (intended for colorectal cancer and glioblastoma) are the biosimilars approved in Canada between 2018 and 2020. Regulatory approval of biosimilars in Canada needs to follow the same process as required by the EMA [132].

### 7.6. Japan

After the USA, Japan is considered to have the second-largest pharmaceutical industry in the world. The regulatory authority of Japan is the Pharmaceutical and Medical Devices Agency (PDMA), which deals with the approval process of biosimilars and published certain guidelines in 2009 based on the EMA guidelines [146]. In contrast to other countries, where the clinical trials must be conducted domestically, in Japan the clinical trials on which the biosimilar approval depends can also be carried out in other Asian countries. Clinical trials are not a compulsory requirement of the PDMA for the approval of biosimilars. Recently, five biosimilars to the reference product trastuzumab (CT-P6, ABP980, PF-05280014, DMB-3111, AP063) have been approved in Japan for BC [147].

### 7.7. South Korea

In the Asia-Pacific region, South Korea is another nation that is well recognized for having lower costs in its pharmaceutical industry. Its regulatory guidelines are based upon the EMA, Japan, and WHO regulations and were standardized in 2009. South Korea was the second country after Europe to provide approval for biosimilars via regulatory pathways. The patent period for biological products in South Korea is 6 years, after which biosimilars can be licensed for the approved reference products. The type of data that is required for the approval process for biosimilars mainly depends on the cases for which their use is intended. Permission for extrapolation of indication can be given to biosimilars when the re-examination period of the reference product for these indications has expired [148].

### 7.8. India

In 2007, after Europe, India was the next country to develop norms and procedures for the approval of biosimilars. Initially, the biosimilars were approved by using the guidelines of the original biological products. In India, the Central Drugs Standard Control Organization (CDSCO) is solely responsible for new medicinal products. In India, the first biosimilar “Biovac-B” was approved for marketing in 2000 for hepatitis B, but no specific guideline was established at that time. However, after approving 20 biosimilars, India also developed regulatory guidelines for the approval of biosimilars [133,149]—in 2012, the Guidelines on Similar Biologics were published, stating the regulatory requirements for marketing authorization in India in association with the Department of Biotechnology; these guidelines were further revised in 2016. Since their revision, the guidelines emphasize post-marketing studies to ensure the safety profile of biosimilars. The reference products for which biosimilars for oncology are being marketed in India include filgrastim, darbepoetin, epoetin, rituximab, and interferon alfa-2b. Clinical trials, which must include assessments of safety and biological equivalence, are the most important criteria for biosimilar licensing in India. For any biosimilar to be licensed in India, its reference product must be approved in India, and approval for extrapolation to any indications cannot be given by the CDSCO [149].

### 7.9. Other Countries

Since 2010, a number of Latin American countries have adopted biosimilar frameworks. Brazil, which has the most established frameworks, has developed two approval pathways for comparable biopharmaceuticals that differ in the evidence required for marketing authorization. In the regulatory framework, comparative studies consisting of phase I pharmacodynamic and pharmacokinetic studies must be further evaluated in phase III clinical trials depending on the type of case. Furthermore, the regulatory approval for the extrapolation of indications with reference to the original products can be given by the respective pathway. Unlike the previous pathway, the individual development pathway requires fewer clinical studies; however, the regulatory approval for extrapolation of indications cannot be permitted. Clinical data from comparative studies are mandatory for the approval of any biosimilar. Moreover, developing countries such as Mexico, Venezuela, and Colombia are also establishing their own regulatory guidelines for the approval of biosimilars [124].

Biosimilars accounts for 40% of the biologics market in China. As cancer has become the country’s second-leading cause of death, the use of biosimilars in oncology has accelerated. However, to date, there are no specific guidelines issued by the China Food and Drug Administration (CFDA). Biosimilars are approved with respect to the regulatory framework of biological products. In the late 1990s, the regulatory guidelines for biopharmaceuticals used in oncology were developed, with a period of 6 years required for any biological product to be marketed [127]. Certain guidelines require large amounts of data associated with preclinical and clinical trials, which has resulted in a decline in applications for new biosimilars. Whether or not a biosimilar can qualify for fast-track drug registration mostly depends on the case and the nature of the reference product. Currently, the CFDA is in the process of developing a regulatory framework for certain biosimilars [127].

Russia has an emerging and firm biopharmaceuticals market but does not have an optimal regulatory framework for biosimilars. There is a need to harmonize regulatory standards for Russian biologics, and for this purpose a meeting was convened by the pharmaceutical manufacturers, international regulatory authorities, and representatives of the Russian Ministry of Health in 2013. Currently, biosimilars are approved through formal legislation defining the regulatory standards with respect to interchangeability and comparative studies. The biosimilars marketed for oncology in the Russian market include rituximab, epoetin, and filgrastim; however, they have not yet been approved by any regulatory authority. The only basis for switching biosimilars is the experience of pharmacists with the appropriate original reference product previously approved in Russia. Rituximab is the only biosimilar that has undergone clinical trials in Russia [127].

For a more vivid description, Table 2 depicts the comparison between the regulatory authorities of different countries, along with the regulatory guidelines associated with pharmacovigilance and the databases used by these countries.

## 8. Adverse Events Associated with the Biosimilars

Safety is a major concern of both patients and practitioners, as biosimilars are widely used in clinical practice for the management and treatment of numerous chronic and life-threatening diseases. These products are associated with low cost in clinical practice, which results in the misconception that biosimilars are not reliable for use in the treatment of life-threatening diseases [155]. According to regulatory guidelines, the approval of marketing of any biosimilar can only be provided if there are “no clinical differences in purity, safety and efficacy” when compared with the original product [156]. Subsequently, the toxicity or side effects associated with biosimilars cannot be distinct from those of the original biopharmaceutical in the post-marketing phase. However, rare and distinctive side effects for any therapeutic indication, if present, can be recognized in comparative studies during human trials [157].

Most biosimilars in oncology are disease-modifying monoclonal antibodies, which suggests that the regulatory consideration is either too expensive or impossible. The regulatory pathways for biosimilars essentially entail the modification of generic pathways for chemically synthesized drugs, but such guidelines have complex characteristics for biopharmaceuticals. The safety profile of biosimilars is directly dependent on their characteristics. However, generally, biopharmaceuticals, biosimilars, and biologics are considered to be safer, as they bind to and target a specific receptor or ligands on the cell membrane. Such agents can be excreted, catabolized, or converted simultaneously into sugars, amino acids, and several other natural products and then recycled. In contrast to small therapeutic molecules, biosimilars tend to not enter the cells and disturb their internal metabolism; as a result, they do not exhibit such toxic effects. However, biopharmaceutical products commonly exert systemic effects instead of pharmacodynamic effects [158].

Biological products are usually administered intravenously; consequently, skin reactions are the most frequent adverse events. Such reactions are usually mild as well as non-specific; however, certain reactions occur due to variation in formulations, and rare side effects can be induced due to immunological and pharmacological effects [8]. The main safety concern associated with biopharmaceutical products is the induction of immunogenicity [159]. It is proposed that all biopharmaceuticals can cause immunogenic adverse events. However, several factors are involved in the occurrence of immunological response, with the main factors being associated with the product. In addition to intrinsic immunogenicity of the protein, product-related causes—including host-cell proteins and aggregation—are primarily responsible for the induction of immunogenicity. However, the intrinsic immunogenicity associated with the original product and the biosimilar will be same, but the differences in physiochemical characteristics may lead to variation in immunogenicity. Particularly, the variation in immunogenic effects between the biosimilar and the original product will be low if there is no significant discrepancy in the physiochemical characteristics of the two [160,161].

Due to the limited number and duration of confirmatory clinical trials, there are equivalent chances of induction of long-term adverse events; however, numerous observational studies and post-marketing surveillance would detect such adverse events. Additionally, no safety signals have been observed even with the use of biosimilars by 400 patients in Europe in clinical practice over more than 10 years. The occurrence of immunogenic effects caused by biosimilars is considered to be their primary safety issue [159]. In addition, biosimilars tend to exhibit these immunogenic effects due to their size and complexity. In addition to manufacturing-related factors and some other factors such as the insertion of host cell proteins, excipients and aggregations may have the ability to induce immunogenicity. As a result, even modest differences between the reference product and the biosimilar can trigger an immunological response—particularly when a patient switches from the original medicine to the biosimilar, or vice versa. However, the possibility of generating an immune reaction still remains even between different batches of the original biological products [161].

This is shown by an incident in Europe involving the production of neutralizing antibodies that cross-reacted with intrinsic erythropoietin, resulting in pure red cell aplasia in approximately 200 patients treated with various epoetin alfa and epoetin beta preparations between 1998 and 2003 [162]. The majority of patients with pure red cell aplasia also had chronic renal disease and were given epoetin subcutaneously. The increase in the prevalence of pure red blood cell aplasia is thought to be attributable to changes in the drugs’ preparation, formulation, handling, and delivery route. Furthermore, after marketing the same biosimilar with improved processes, the occurrence decreased by more than 80%. A thorough assessment of immunological reactions to biosimilars is an important element of the development process, and a candidate biosimilar’s safety profile is also essential for approval in both Europe and the United States.

Additionally, numerous clinical trials on biosimilars are also conducted to ensure that there is no unexpected immunogenicity by switching a reference product to a biosimilar, or vice versa. For instance, clinical trials of biosimilars for chronic inflammatory diseases (i.e., etanercept-szzs, infliximab-abda, adalimumab-atto, and infliximab-dyyb) switched patients from the original biological product to a biosimilar, and no considerable differences in immunogenicity were observed [163,164,165,166]. Numerous switches between biosimilars and reference products would be expected to evaluate the potential for immune reaction, and many trials can be conducted to analyze immunogenicity more precisely. There are six treatment cycles associated with the biosimilar filgrastim-sndz for FDA approval, carried out in phase III PIONEER trials in patients with BC receiving myelosuppressive CHM. This population received alternative cycles of treatment with the reference product and the biosimilar, and no significant differences in the safety and efficacy were observed between the drugs [167].

Since the number of clinical studies required for regulatory clearance of biosimilars is quite low, regulatory bodies such as the FDA and EMA rely heavily on post-marketing surveillance to assess the safety profile of biosimilars. However, the current pharmacovigilance system for biosimilars is not considered to be an appropriate source for reporting all of the adverse events. This is due to the lack of time and resources available to the healthcare professionals. The utmost challenge for the physicians is the increasing administrative burden, which can lead to incomplete electronic health records of the patients. Many systems are currently utilized by various regulatory authorities to keep the records of adverse events, such as the FDA’s Sentinel initiative, EudraVigilance, post-marketing safety registries, and integrated healthcare information systems such as ASCO CancerLinQ. Such software programs are copious sources of significant information about the safety and efficacy of biosimilars [5]. There are numerous adverse events associated with biosimilars. Such adverse events can be rare, common, or uncommon, and some of them occur at an unknown frequency. However, there are many biosimilars that are approved by the FDA, and Figure 9 shows the adverse drug reactions associated with these biosimilars.

## 9. Traceability of Biosimilars in the Post-Marketing Phase

The available pre-marketing data on the clinical safety and efficacy of biosimilars are limited; consequently, the benefit–risk profiles of these agents are frequently questioned by oncologists. However, the approval of biosimilars requires comparative studies with their original biological products. Moreover, the safety profiles of biosimilars are identical to those of their original biological products, indicating that biosimilars and biologics are subject to similar regulatory aspects of pharmacovigilance in many countries [167]. Furthermore, it is mandatory for both biologics and biosimilars to be monitored particularly closely in pediatrics and geriatrics in both the clinical and post-marketing phases, as the comorbidity profiles of children and the elderly can differ from those of the adults who participate in clinical trials [168].

However, specific risk management strategies are required for biopharmaceuticals, as current post-marketing safety parameters for both chemically synthesized molecules and biopharmaceuticals are comparable. Pragmatic clinical trials, post-marketing observational studies, and spontaneous reporting systems are the main sources for the collection of safety information about biosimilars. The spontaneous reporting system acts as the imperative tool for the identification of potential safety signals, as well as the significance of immunogenicity—particularly if less severe—although additional development is still required in certain tools for the detection of potential safety issues. Furthermore, case-causality assessment in adverse drug reaction reports during spontaneous reporting is another complication related to biopharmaceuticals. Patients treated with biologics are usually on polytherapy and suffer from severe or life-threatening disorders, making the causality assessment more complicated [89]. Another complication in the case-causality assessment of adverse events associated with biopharmaceuticals is channeling bias, in which a disease state is inappropriately attributed to the use of the drug, and further such drugs are prescribed to the population who are more likely to develop the adverse reactions.

Another challenge associated with pharmacovigilance of biopharmaceuticals is the variability in synthesis over certain periods of time in clinical practice. Subsequently, an important requirement is to safeguard the product and batch traceability of biopharmaceuticals in the post-marketing phase. According to regulatory agencies, healthcare professionals have to specify the brand name as well as the batch number in spontaneous case reports for the better identification of biopharmaceutical products, but this requirement is not completely adopted. Other than RMP, which is based on European Pharmacovigilance legislation, supplementary post-marketing surveillance and many other phase IV clinical studies are required for the evaluation of biosimilars and biologics. Moreover, in the USA, a Risk Evaluation and Mitigation Strategy is a mandatory requirement for manufacturers in order to analyze the safety profile of biologics and biosimilars, where the benefits must outweigh the risks [169].

Variations in the manufacturing process of biopharmaceuticals occur frequently when these agents are marketed. Usually, these variations do not have any negative impact on the safety and efficacy profile of biopharmaceuticals; however, the instance of epoetin-induced pure red cell aplasia demonstrated that minor alterations in formulation can potentially lead to the occurrence of adverse events. As a result, continuous monitoring of biologics and biosimilars is a mandatory requirement for pharmacovigilance in order to safeguard the marketed product and the traceability of multiple batches marketed and prescribed in clinical practice. For any biopharmaceutical that has been marketed, the specific product and batch administered to the patient should be traced so that all of the adverse events can be detected and analyzed optimally.

The pharmacovigilance legislation expressed in Directive 2010/84/EU, adopted by the European Parliament and Council of Ministers, declared that member states must ensure that all adequate methods should be taken to identify biologics with suspected ADRs, focusing on brand name as well as batch number [90]. A previous study performed by Vermeer et al. (2013) reported that 24% and 21.1% of spontaneous ADR reporting for biological products is carried out via the FDA’s Adverse Event Reporting System and the European Union’s EudraVigilance, respectively, using information on batch number [91]. Moreover, this study also showed that the prescribed biopharmaceuticals could be traced back to the manufacturer by brand name in 96.2% of the cases, while only 5.7% of the cases had information about the batch number. Additionally, the spontaneous reports submitted by consumers contain the batch number more often than the reports submitted by pharmacists and clinicians. Recently, a study on the Italian spontaneous reporting system showed similar results, stating that 94.8% of biologic-related reports contained information about brand name, while only 8.6% of case reports contained the batch number [91].

The main limitation for any biopharmaceutical product in the market is that while the product name and the batch number are mentioned on the product packaging, batch numbers are not mentioned in the barcodes of particular biological products. In order to ensure the traceability of biopharmaceuticals, accurate records and data associated with brand name and batch number must be recorded at all levels [170]. In 2020, Klein et al. [171] also reported similar results in the identification of biological products in Dutch hospitals. Data revealed that recording of the brand names of such agents is well established, but batch numbers are poorly recorded; 76% of ADR reports associated with recombinant biologics had a traceable brand name, while only 5% of ADR reports contained a batch number. As a result, a huge improvement is required in the traceability standards of biopharmaceutical products, including both biologics and biosimilars—especially the reporting of batch numbers in spontaneous reports, which will also allow the better identification and assessment of the safety monitoring profiles of drugs in clinical practice [171].

## 10. Conclusions

Biopharmaceutical products have widely been used as potential treatments and for palliative care in the treatment of cancers and tumors. Despite being targeted agents, biological products are higher in cost, which exacerbates the global crises in healthcare systems. Moreover, patent expiry of such biological products has facilitated the marketing of biosimilars that are lower in cost and have similar efficacy and safety profiles to the original biologics. Therefore, the treatment of cancer can be accessed by the majority of the population, with biosimilars providing novel treatments for various life-threatening and chronic diseases. Patients who were administered an infliximab biosimilar ultimately spent 12% less out of pocket than with the reference biologic, according to a Johns Hopkins study [172]. The price comparison between biologics and biosimilars in the study showed that, when all attributes were taken into account, the biosimilars’ price represented 68% of the cost of biologics for infliximab and 74% of the cost of biologics for filgrastim [172].

Currently, there are many countries that are still developing regulatory frameworks for the approval and licensing of biosimilars. However, more than 10 years of experience in utilizing biosimilars in places such as Europe, Canada, the USA, and Australia provides an optimistic depiction of the safety profile of biosimilars. There are many challenges associated with the underuse of biosimilars, in addition to the innumerable uses of biosimilars in oncology. Many oncologists are not aware of the advantages of using biosimilars over biologics, due to lack of safety data available in the market. Hence, there is a need to analyze and evaluate the safety profiles of biosimilars so that clinicians can become more aware of the benefits of utilizing biosimilars in clinical practice. More frequent monitoring in the clinical and post-marketing phases is required in order to ensure that these new products are safe and effective in real-world settings. Furthermore, clinicians and healthcare professionals, as well as patients, should be educated optimistically about the outlook of utilizing biosimilars in order to ensure their successful incorporation in routine oncological care.

## 11. Future Perspectives

There are many medicinal products on the market that have been utilized for the management and treatment of different cancers and tumors. Some of the traditional medications involve CHM and monoclonal antibodies, but there is a need for more targeted therapies in order to treat cancer more precisely. Such targeted therapies comprise biopharmaceutical products and biologics that have been used in the field of oncology since the 1990s. However, biologics are higher in cost; consequently, there is a large population who do not receive the proper treatment, which contributes to higher fatalities due to cancer and tumors. This can be eliminated with the more frequent use of biosimilars, which are similar to the biologics in terms of safety and efficacy. The use of biosimilars by oncologists can reduce the burden of treatment among the population, while also reducing the risk of mortality due to cancer and tumors. Despite the approval of many biosimilars in the market, there are limited amounts of studies that have been conducted in the post-approval phase of the biosimilars.

Biosimilars have the potential to save a lot of money for the health sector, but only if they are adopted and used in clinical practice. It can be stated that doctors are still skeptical of these biosimilar treatments, in part due to a lack of familiarity. Biosimilars are important for the future of healthcare because they boost market competition and innovation, leading to reduced prices and increased pharmaceutical availability for patients around the world. Clinicians, on the other hand, are wary about using biosimilars and do not generally endorse their usage as safe and effective treatment alternatives in patients who are already receiving bio-originator therapy. Even in the published literature, biosimilar-specific education appears to be a somewhat ignored area of focus. As a result, there is a critical requirement to raise healthcare professionals’ awareness and confidence. Additionally, educational programs should concentrate on the major areas of immunogenicity, extrapolation, and interchangeability. To develop continuous evidence-based education programs for biosimilars, efforts and incentives should be implemented. This would contribute to better knowledge of biosimilars on all levels, including clinical, molecular, and regulatory. Furthermore, to improve the knowledge of healthcare professionals, the central regulatory agencies that are formally in charge of ensuring the quality, safety, and efficacy of medicines should adopt the following points to demonstrate interchangeability when applying for approval: (a) change the paradigm for biosimilar development; (b) grant interchangeability automatically upon approval; (c) publish a position statement strongly endorsing interchangeability as a first-course initiative; and (d) harmonize the fundamental ideas of interchangeability, switching, and substitution.

## Figures and Tables

**Figure 1 pharmaceutics-14-02721-f001:**
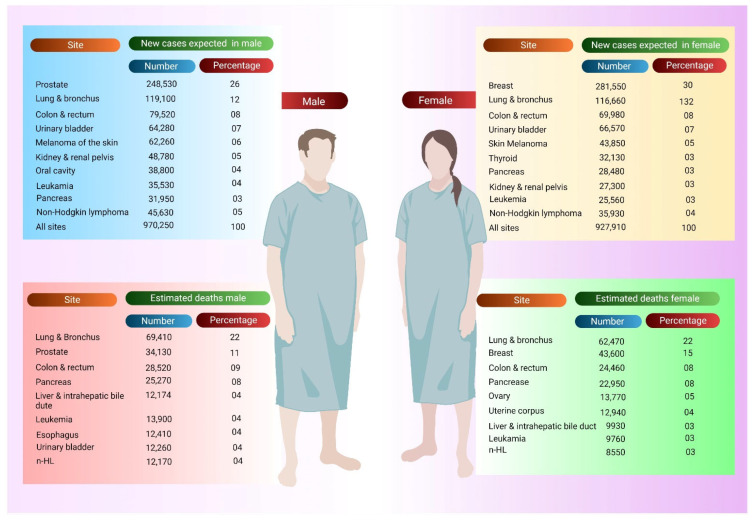
New cancer cases and deaths in the major sites—estimates for 2021 [18].

**Figure 2 pharmaceutics-14-02721-f002:**
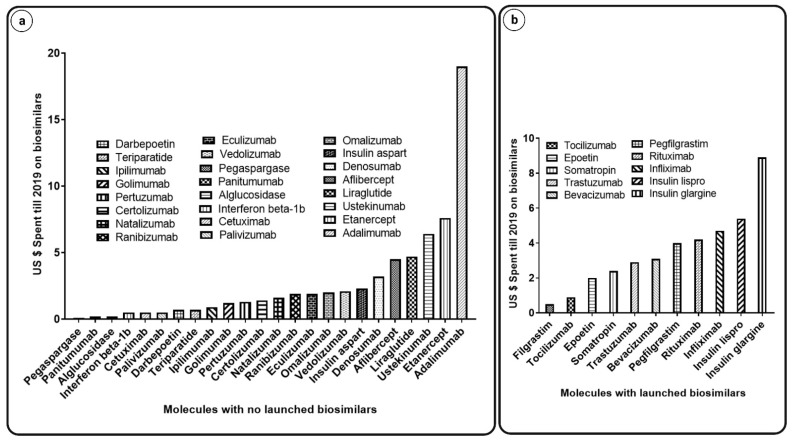
USD spent until 2019 on biosimilars; Graphs indicating (**a**) molecules with no launched biosimilars versus USD spent till 2019 on biosimilars and (**b**) molecules with launched biosimilars versus USD spent till 2019 on biosimilars [22].

**Figure 3 pharmaceutics-14-02721-f003:**
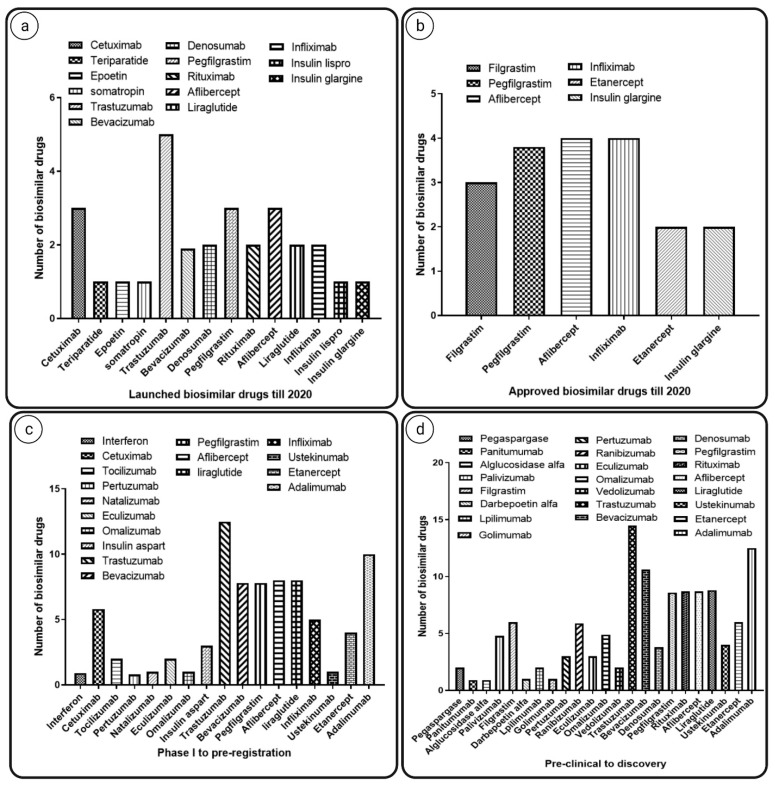
Biosimilars launched, approved, or in development. The graphs indicate number of (**a**) biosimilar drugs launched till 2020; (**b**) approved biosimilars till 2020; (**c**) biosimilar that were in phase 1 clinical trial to pre-registration; (**d**) biosimilars that were in their pre-clinical phase to discovery [22].

**Figure 4 pharmaceutics-14-02721-f004:**
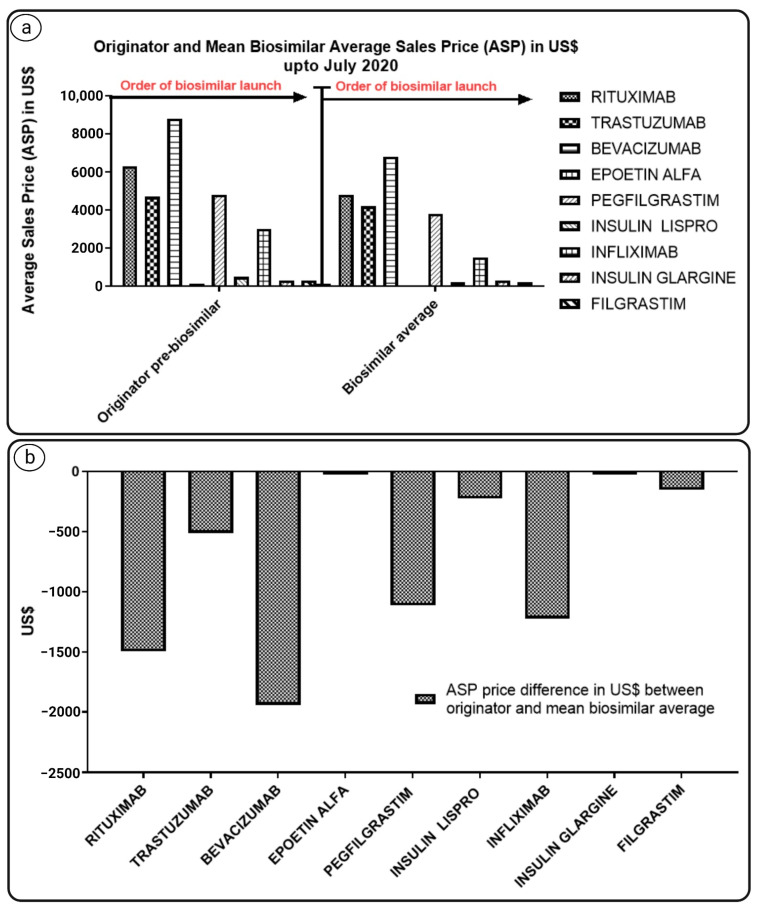
The graph indicates (**a**) average sales prices (ASPs) in USD of originators and mean biosimilar up to July 2020; (**b**) difference in ASP in USD of originators and mean biosimilar [22].

**Figure 9 pharmaceutics-14-02721-f009:**
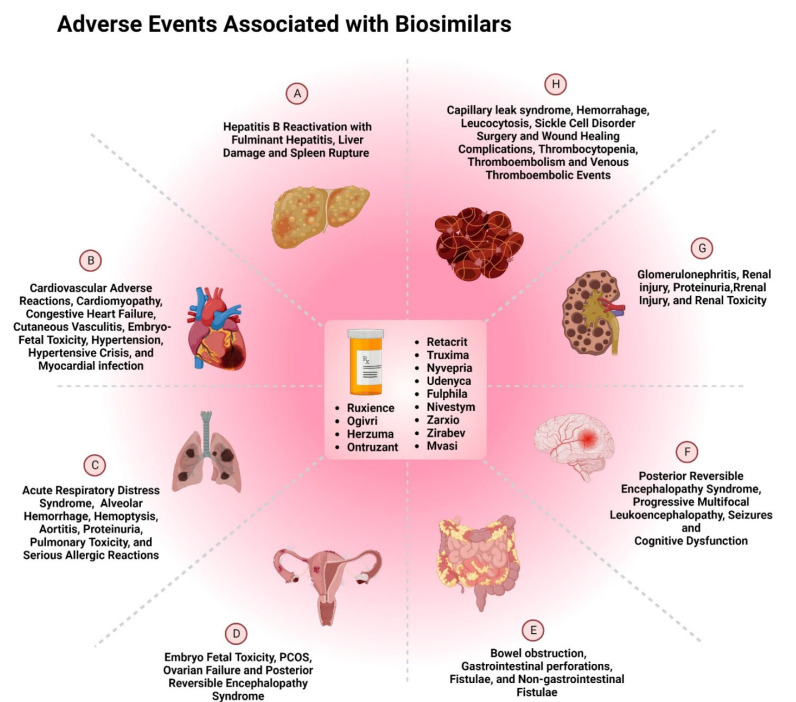
Adverse events associated with FDA-approved biosimilars used in treatment of A. liver; B. cardiovascular; C. Lung; D. Uterine and cervical; E. Gastrointestinal; F. Brain; G. Kidneys; and H. Circulatory System [42,52,58,59,60,66,67,69,73,74,75,76,77,79].

**Table 1 pharmaceutics-14-02721-t001:** Recently approved biosimilars in oncology.

Active Ingredient	Year	Biosimilars	Manufacturer	Country/Regulatory Body	Therapeutic Areas	References
Epoetin alfa	2007	Abseamed	Medice Arzneimittel Putter GmbH & Co. Kg	Europe/EMA	Cancer, ANE, CKF	[36]
	2007	Binocrit	Sandoz GmbH	Europe/EMA	ANE, CKF	[37]
	2007	Epoetin-α Hexal	Hexal Ag	Europe/EMA	ANE, cancer, ANE, CKF	[38]
	2007	Retacrit	Hospira UK Limited	Europe/EMA	ANE, cancer, ABT, CKF	[39]
	2007	Silapo	Stada Arzneimittel Ag	Europe/EMA	ANE, cancer, ABT, CKF	[40]
Epoetin lambda	2010	Novicrit	Sandoz	Australia/TGA	ANE, cancer, CKF	[41]
	2010	Aczicrit	Sandoz	Australia/TGA	ANE, cancer, CKF	
	2010	Grandicrit	Sandoz	Australia/TGA	ANE, cancer, CKF	
Epoetin alfa-epbx	2018	Retacrit	Hospira Inc.	USA/FDA	ANE (caused by chronic kidney disease, CHM, or patients taking zidovudine), reduction in allogeneic red blood cell transfusions	[42]
Filgrastim	2008	Ratiograstim	Ratiopharm GmbH	Europe/EMA	Cancer, hematopoietic stem cell transplantation (HSCT), NTP	[43]
	2008	Tevagrastim	Teva GmbH	Europe/EMA	Cancer, HSCT, NTP	[44]
	2009	Zarzio	Sandoz	Europe/EMA	Cancer, HSCT, NTP	[45]
	2009	Filgrastimhexal	Hexal Ag	Europe/EMA	Cancer, HSCT, NTP	[46]
	2010	Nivestim	Hospira UK	Europe/EMA	Cancer, HSCT, NTP	[47]
	2010	Nivestim	Hospira	Australia/TGA	Cancer, HSCT, NTP	[41]
	2011	Tevagrastim	Aspen Pharmacare Australia	Australia/TGA	Cancer, HSCT, NTP	[41]
	2013	Zarzio	Sandoz	Australia/TGA	Cancer, HSCT, NTP	[41]
	2013	Grastofil	Apotex Europe Bv	Europe/EMA	NTP	[48]
	2014	Accofil	Accord Healthcare Ltd.	Europe/EMA	NTP	[49]
	2015	Zarxio	Sandoz Inc.	USA/FDA	BMT, APBPC, NTP, ML	[50]
	2015	Grastofil	Apotex	Canada/Health Canada	NTP	[51]
Filgrastim-aafi	2018	Nivestym	Pfizer Inc.	USA/FDA	BMT, APBPC, cancer, NTP, ML	[52]
*Pegfilgrastim*	2018	Udenyca	Coherus	Europe/EMA	NTP	[53]
	2018	Fulphila	Mylan S.A.S.	Europe/EMA	NTP	[54]
	2018	Pelmeg	Cinfa Biotech S.L.	Europe/EMA	NTP	[55]
	2018	Ziextenzo	Sandoz GmbH	Europe/EMA	NTP	[56]
	2018	Ogivri	Mylan S.A.S.	Europe/EMA	MBC, EBC, MGC	[57]
	2020	Ziextenzo	Sandoz	Australia/TGA	CHM-induced NTP, HSCT	[41]
	2020	Fulphila	Alphapharm	Australia/TGA	CHM-induced NTP, HSCT	
	2020	Pelgraz	Accord	Australia/TGA	CHM-induced NTP, HSCT	
	2020	Ziextenzo	Sandoz	Canada/Health Canada	NTP	[51]
Pegfligrastim-jmdb	2018	Fulphila	Mylan/Biocon	USA/FDA	FN	[58]
Pegfligrastim-cbqv	2018	Udenyca	Coherus Biosciences	USA/FDA	FN	[59]
Pegfilgrastim-apgf	2020	Nyvepria	Pfizer Inc.	USA/FDA	FN	[60]
Rituximab	2017	Riximyo	Sandoz Australia	Australia/TGA	Chronic lymphocytic leukemia (CLL), B-cell NHL, MPA, WG, RA	[41]
	2017	Truxima	Celltrion Healthcare Hungary Kft	Europe/EMA	GPA, CLL, MPA, RA, n-HL	[61]
	2017	Riximyo	Sandoz GmbH	Europe/EMA	B-CLL, n-HL, MPA, WG, RA	[62]
	2017	Rixathon	Sandoz GmbH	Europe/EMA	MPA, CLL, B-cell n-HL, WG, RA	[63]
	2017	Ritemvia	Celltrion Healthcare Hungary Kft.	Europe/EMA	WG, n-HL, MPA	[64]
	2017	Blitzima	Celltrion Healthcare Hungary Kft.	Europe/EMA	B-CLL, n-HL	[65]
	2018	Truxima	Celltrion	Australia/TGA	B-cell NHL, MPA, CLL, WG, RA	[41]
	2020	Riximyo	Sandoz	Canada/Health Canada	RA, n-HL, CLL	[51]
	2020	Ruxience	Pfizer	Canada/Health Canada	RA, n-HL, CLL	
Rituximab-abbs	2018	Truxima	Celltrion Inc.	USA/FDA	n-HL	[66]
Rituximab-pvvr	2019	Ruxience	Pfizer Inc.	USA/FDA	CLL, GPA, n-HL	[67]
Trastuzumab	2017	Ontruzant	Samsung Bioepis	Europe/EMA	EBC, MGC, MBC	[68]
	2017	Ogivri	Mylan	USA/FDA	HER2 BC, AC, HER2 MGGJ,	[69]
	2018	Herzuma	Celltrion Healthcare	Australia/TGA	EBC, MGC, MBC	[41]
	2018	Herzuma	Celltrion Healthcare Hungary Kft.	Europe/EMA	EBC, MGC, MBC	[70]
	2018	Kanjinti	Amgen Europe	Europe/EMA	EBC, MGC, MBC	[71]
	2018	Trazimera	Pfizer Europe	Europe/EMA	Stomach neoplasms, breast neoplasms	[72]
	2019	Herzuma	Celltrion	Canada/Health Canada	BC, GC	[51]
	2019	Ogivri	Alphapharm	Australia/TGA	BC, GC	[41]
	2019	Kanjinti	Amgen	Australia/TGA	BC, GC	
	2020	Ontruzant	Merck Sharp and Dohme	Australia/TGA	BC, GC	
	2020	Trazimera	Pfizer	Australia/TGA	BC, GC	
	2020	Kanjinti	Amgen Canada	Canada/Health Canada	EBC, MGC, MBC	[51]
Trastuzumab-pkrb	2018	Herzuma	Celltrion Inc.	USA/FDA	BC	[73]
Trastuzumab-dttb	2019	Ontruzant	Samsung Bioepis	USA/FDA	HER2 BC, HER2 MGGJ, AC	[74]
Trastuzumab-qyyp	2019	Trazimera	Pfizer Ind.	USA/FDA	HER2 BC, HER2 MGGJ, AC	[75]
Trastuzumab-anns	2019	Kanjinti	Amgen Inc.	USA/FDA	HER2 BC, HER2 MGGJ, AC	[76]
Bevacizumab	2017	Mvasi	Amgen, Inc.	USA/FDA	NSCLC, RCC, colorectal neoplasms, breast neoplasms, ovarian neoplasms	[77]
	2018	Mvasi	Amgen Europe B.V.	Europe/EMA	Breast neoplasms, non-small-cell lung carcinoma, fallopian tube neoplasms, peritoneal neoplasms, ovarian neoplasms, RCC	[78]
Bevacizumab-bvzr	2019	Zirabev	Pfizer Inc.	USA/FDA	NSCLC, CRC, cervical cancer, breast neoplasms, RCC	[79]

Key: ABT = autologous blood transfusion; AC = adenocarcinoma; ANE = anemia; APBPC = autologous peripheral blood progenitor cell; B-CLL = B-cell chronic lymphocytic leukemia; BMT = bone marrow transplantation; CKF = chronic kidney failure; CLL = chronic lymphocytic leukemia; CRC = colorectal cancer; EBC = early breast cancer; EMA = European Medicines Agency; FDA = Food and Drug Administration; FN = febrile neutropenia; GC = gastric cancer; GPA = granulomatosis with polyangiitis; HER2 = human epidermal growth factor receptor 2; MBC = metastatic breast cancer; MGC = metastatic gastric cancer; MGGJ = metastatic gastric or gastroesophageal junction; ML = myeloid leukemia; MPA = microscopic polyangiitis; NTP = neutropenia; n-HL = non-Hodgkin lymphoma; NSCLC = non-small-cell lung cancer; RCC = renal cell carcinoma; TGA = therapeutic goods administration; USA = United States of America; WG = Wegener granulomatosis.

**Table 2 pharmaceutics-14-02721-t002:** Comparison between the guidelines of various countries.

Country	Regulatory Bodies	Regulatory Guidelines	Authorities Responsible for Pharmacovigilance	Database	Reference
Europe	EMA	European Medicines Agency: Similar Biological Medicinal Products (overarching guidelines). CHMP/437/04 Rev. 1 European Medicines Agency: Guideline on Good Pharmacovigilance Practices (GVPs), Product- or Population-Specific considerations II: Biological Medicinal Products. EMA/168402/2014	European Commission, EudraVigilance Data Analysis System (EVDAS), Good Pharmacovigilance Practices (GVP)	European Database of Suspected Adverse Drug Reaction Reports, European Commission, EudraVigilance Data Analysis System (EVDAS), EudraVigilance Web Trader (EVWEB)	[150]
Canada	Biologics and Genetics Therapy Directorate (BGTD), Health Products and Food Branch (HPFB)	Information and Submission Requirements for Biosimilar Biologic Drugs (2010, revised 2016)	Health Canada	Vigilance	[151]
USA	FDA	Labeling for Biosimilar Products; Guidance for Industry (2016) Reference Product Exclusivity for Biological Products Filed Under Section 351(a) of the PHS Act (2015)	Center for Drug Evaluation and Research (CDER), Center for Biologics Evaluation and Research (CBER)	FAERS, Sentinel System	[152]
Australia	Therapeutic Goods Administration	Pharmacovigilance Responsibilities of Medicine Sponsors: Australian Recommendations and Requirements. Risk Management Plans for Prescription Medicines	Therapeutic Goods Administration	Database of Adverse Event Notification (DAEN)	[153]
South Korea	Korean Food and Drug Administration	Guideline on Evaluation of Biosimilar Products Regulation on Marketing Authorization of Biopharmaceutical Products (2007)	Ministry of Food and Drug Safety	Korea Adverse Event Reporting System (KAERS)	[154]

Key: BGTD = Biologics and Genetics Therapy Directorate; CBER = Center for Biologics Evaluation and Research; CDER = Center for Drug Evaluation and Research; CHMP = Committee for Medicinal Products for Human Use; DAEN = Database of Adverse Event Notification; EVDAS = EudraVigilance Data Analysis System; EVWEB = EudraVigilance Web Trader; FAERS = FDA Adverse Event Reporting System; FDA = Food and Drug Administration; GVP = Good Pharmacovigilance Practices; HPFB = Health Products and Food Branch; KAERS = Korea Adverse Event Reporting System.

## Data Availability

Not applicable.

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
