# Peer review of "Biosimilars in Oncology: Latest Trends and Regulatory Status"

_pharmaceutics, 2022, doi:10.3390/pharmaceutics14122721_

Round 1

Reviewer 1 Report

Review submitted October 4, 2022

Biosimilars in Oncology: Latest Trends and Regulatory Status

This is a timely and very comprehensive review of the global biosimilars landscape. It compares the regulatory guidelines of biosimilars across the world, and also presents the latest trends and challenges in medical oncology both now and in the future which will assist healthcare professionals, payers, and patients in making informed decisions, increasing biosimilar acceptability in clinical practice, increasing accessibility, and speeding up the health and economic benefits associated with biosimilars.

This review should be published in “Pharmaceutics”, pending a minor revision. I have a few comments, queries and concerns I believe the authors should address before their excellent review will be published.

1) Beginning on line 111, the authors write: “in 2016, 89.9% of surveyed clinicians op-111 posed pharmacy-level substitution of a biosimilar, and then in another 2016 survey, 80% 112 of survey participants were unaware that an interchangeability designation could result 113 in automatic substitution [16]”.

These are not global statistics as one may think: the survey of reference 16 reported the trends among USA and European Clinicians. This should be mentioned.

2) From where did the data shown in Figure 1 originate? What is the reference or references?

3) There is a problem with the way data are presented in Figure 2: on the left column, “molecules with no launched biosimilars” are colored blue. However, there are biosimilars for Adalimumab (listed on top), as evident from the right column (where Adalimumab is on top too) and from the following text. Please check carefully how molecules are color-coded in Figure 2!

4) Beginning on line 289: “Biosimilars might save the 289 US between $67 and $108 billion between 2010 and 2019, according to predictions made 290 in 2008”.

What is the Reference for this estimation?

The estimation was for the period 2010-2019. Now we are in 2022. Can the authors check of the prediction was fulfilled?

5) Beginning on line 331: “EMA is the ungenerous regulatory authority . . .”

What is the need to label the EMA as “ungenerous”?

6) Beginning on line 381: Rituximab is defined as “It is a genetically engineered chimeric monoclonal antibody which tends to restrain the antigen CD20 from binding to its antibodies”

This is a very strange and inaccurate definition of Rituximab

7) Beginning on line 667: “These products are associated with low cost in the clinical practice which results in the misconception that Biosimilars are not much reliable to be used in life-threatening diseases”

This is a strange statement – what evidence is it based on? Is there a reference that lists this misconception as common among healthcare professionals?

8) Beginning on line 718: “This is shown by an incident in Europe that involves the production of neutralising antibodies that cross-reacted with intrinsic erythropoietin, resulting in pure red cell aplasia in approximately 200 patients cured with various epoetin alfa and epoetin beta preparations”

I have a reservation about using the word “cured”. I assume “treated” will more accurately define what the patients underwent.

9) Beginning on line 755: the authors write: “frequency. The adverse events of currently marketed Biosimilars in Europe is available as a supplementary material”

No “supplementary material” was presented to my evaluation with the manuscript.

10) Since, as I wrote in comment 9, there is no supplementary material. I ask that the data shown in Figure 8 will be supported by References.

11) In part 9. Traceability of Biosimilars in post-marketing phase

I appreciate the authors concern with too-low indication of batch numbers on marketed biosimilars. If this is the case, it should be rectified. However, is this practice different from the labeling of originator drugs? I don’t think so. In my opinion, biosimilars should not be subjected to higher standards of scrutiny than originator biopharmaceuticals. Therefore, I will appreciate seeing a comment that better Traceability standards should be applied to both (relevant to what they wrote beginning on line 833).

12) In the conclusion, beginning on line 844, the authors relate to the effect of the reduced prices of biosimilars in writing: “Therefore, the treatment of cancer can be accessed by majority of population which makes the Biosimilars a novel treatment for the treatment of various life-threatening and chronic diseases”

This, indeed, makes sense. Can the authors cite an example (or examples) where an originator drug was withheld from patients due to high cost and once cheaper biosimilars were approved they penetrated into that market (country)?

13) Many of the reference contain URLs which are not accessible due to being truncated by the line number at the end of the line. This is a comment to the editorial office: please make sure that all the URLs will be viable in the final version that will be published.

Author Response

1) Beginning on line 111, the authors write: “in 2016, 89.9% of surveyed clinicians op-111 posed pharmacy-level substitution of a biosimilar, and then in another 2016 survey, 80% 112 of survey participants were unaware that an interchangeability designation could result 113 in automatic substitution [16]”.

These are not global statistics as one may think: the survey of reference 16 reported the trends among USA and European Clinicians. This should be mentioned.

Response: As suggested the countries are included to remove the confusion.

2) From where did the data shown in Figure 1 originate? What is the reference or references?

Response: Reference is added with Figure 1 now.

3) There is a problem with the way data are presented in Figure 2: on the left column, “molecules with no launched biosimilars” are colored blue. However, there are biosimilars for Adalimumab (listed on top), as evident from the right column (where Adalimumab is on top too) and from the following text. Please check carefully how molecules are color-coded in Figure 2!

Response: We have not given any colour to the bar diagrams. I would request the learned reviewer to clarify it so that we can be able to address this comment.

4) Beginning on line 289: “Biosimilars might save the 289 US between $67 and $108 billion between 2010 and 2019, according to predictions made 290 in 2008”.

What is the Reference for this estimation?

Response: Latest data with reference is added.

The estimation was for the period 2010-2019. Now we are in 2022. Can the authors check of the prediction was fulfilled?

 Response: Latest data with estimation period 2021-2025 is added.

5) Beginning on line 331: “EMA is the ungenerous regulatory authority . . .”

What is the need to label the EMA as “ungenerous”?

Response: The “ungenerous” word was used to show the unbiased framework of EMA for the scientific evaluation, supervision and safety monitoring of medicines in the EU and that’s why other countries also follows the same framework. However, there is no harm in removing the word, therefore I have rephrased the sentence now for better clarity.

6) Beginning on line 381: Rituximab is defined as “It is a genetically engineered chimeric monoclonal antibody which tends to restrain the antigen CD20 from binding to its antibodies”

This is a very strange and inaccurate definition of Rituximab

Response: The definition is now elaborated to give the accurate description of the product.

7) Beginning on line 667: “These products are associated with low cost in the clinical practice which results in the misconception that Biosimilars are not much reliable to be used in life-threatening diseases”

This is a strange statement – what evidence is it based on? Is there a reference that lists this misconception as common among healthcare professionals?

Response: A systematic review focussing on this misconception is added as a reference.

8) Beginning on line 718: “This is shown by an incident in Europe that involves the production of neutralising antibodies that cross-reacted with intrinsic erythropoietin, resulting in pure red cell aplasia in approximately 200 patients cured with various epoetin alfa and epoetin beta preparations”

I have a reservation about using the word “cured”. I assume “treated” will more accurately define what the patients underwent.

Response: We have accepted your suggestion. Thank you.

9) Beginning on line 755: the authors write: “frequency. The adverse events of currently marketed Biosimilars in Europe is available as a supplementary material”

 No “supplementary material” was presented to my evaluation with the manuscript.

Response: This sentence was mistakenly written. We have deleted it.

10) Since, as I wrote in comment 9, there is no supplementary material. I ask that the data shown in Figure 8 will be supported by References.

Response: I think you are referring Figure 9 in your comment. We have added the references now.

11) In part 9. Traceability of Biosimilars in post-marketing phase

I appreciate the authors concern with too-low indication of batch numbers on marketed biosimilars. If this is the case, it should be rectified. However, is this practice different from the labeling of originator drugs? I don’t think so. In my opinion, biosimilars should not be subjected to higher standards of scrutiny than originator biopharmaceuticals. Therefore, I will appreciate seeing a comment that better Traceability standards should be applied to both (relevant to what they wrote beginning on line 833).

Response: We completely agree with your opinion, the requirement of better traceability standards is applicable for all the biopharmaceutical products (including biologics and biosimilars) and this has been already included under this section. Thank you.

12) In the conclusion, beginning on line 844, the authors relate to the effect of the reduced prices of biosimilars in writing: “Therefore, the treatment of cancer can be accessed by majority of population which makes the Biosimilars a novel treatment for the treatment of various life-threatening and chronic diseases”

This, indeed, makes sense. Can the authors cite an example (or examples) where an originator drug was withheld from patients due to high cost and once cheaper biosimilars were approved they penetrated into that market (country)?

Response: Examples added to show the saving opportunities available with Biosimilars. 

13) Many of the reference contain URLs which are not accessible due to being truncated by the line number at the end of the line. This is a comment to the editorial office: please make sure that all the URLs will be viable in the final version that will be published.

Response: Comment for editorial office

Reviewer 2 Report

Except English language editing, this study is good to be published, especially it is of interest to the local readership.

Author Response

Except English language editing, this study is good to be published, especially it is of interest to the local readership.

Response: As per the suggestions of learned reviewer we have improved the language of manuscript at relevant positions

Reviewer 3 Report

The issue is interesting; however, the paper has some important flaws.

The text is often prolix and must be shortened; some statements are too general.

FOR EXAMPLE, in the Introduction section:

Lines 64-65 “diseases as well as different types of cancers and tumors have been reconfigured by the biologics” this statement is generical and inaccurate

Line 66 “cancers and tumors” this expression could be replaced by “malignancies”

Lines 67-68 “However, the cost as … development of ‘Biosimilars’ [3, 4]”. This phrase is unclear.

Lines 80-82 “Consequently, European Medicine Agency (EMA) … the Europe in 2006 [11]” this issue is not related to cancer.

Lines 83-84 “It has been observed … and medication of cancer [12]” This phrase is generical and unclear.

Lines 85-86 “In 2015, FDA introduced … in USA in 2015” this phrase must be re-written.

Line 93 “have not got recognized” please correct

Line 95 “immunogenicity or immunoreactivity occurs when the body's immune system recognizes the biomaterial as a foreign item” this is useless.

Lines 105-106 “but still, switching … in terms of safety” this phrase is redundant.

Lines 115-130 “This article's main goal … and economic benefits”. This is too long and useless.

Moreover:

The paragraphs 2.2. Risk factors and 2.3. Treatment are useless.

Figures 2, 3, 4 must be limited to drugs used in oncology.

The paper must be shortened and re-organized: I suggest to discuss first the medical aspects (paragraphs 5 and 8), then the economic and regulatory aspects.

Some figures and tables (i.e. after parafraph 6.6, or in paragraph 8) are not defined. Figure 9 seems to be useless.

English language needs an extensive revision

Author Response

1. The issue is interesting; however, the paper has some important flaws.

The text is often prolix and must be shortened; some statements are too general. FOR EXAMPLE, in the Introduction section:

Lines 64-65 “diseases as well as different types of cancers and tumors have been reconfigured by the biologics” this statement is generical and inaccurate

Response: As it is well known fact that the Biologics have revolutionized patient management in multiple disease states, therefore we have rephrased the statement to give the clear picture.

2. Line 66 “cancers and tumors” this expression could be replaced by “malignancies”

Response: The words are replaced based on your suggestion. Thank you.

3. Lines 67-68 “However, the cost as … development of ‘Biosimilars’ [3, 4]”. This phrase is unclear.

Response: Statement is rephrased for better clarity.

4. Lines 80-82 “Consequently, European Medicine Agency (EMA) … the Europe in 2006 [11]” this issue is not related to cancer.

Response: The product ‘Omnitrope’ is removed, however the information related to biosimilar guidelines is retained as it was applicable for all types of biosimilars including oncology.

5. Lines 83-84 “It has been observed … and medication of cancer [12]” This phrase is generical and unclear.

Response: Yes, the statement is general and applicable to all the Biosimilars but as it shows the increase in percentage of treatment compliance after the introduction of Biosimilars, we think it can be retained. However, for better clarity we have rephrased the statement.

6. Lines 85-86 “In 2015, FDA introduced … in USA in 2015” this phrase must be re-written.

Response: The sentences are separated by period ‘.’ to avoid confusion.

7. Line 93 “have not got recognized” please correct

Response: Corrected now. Thank you.

8. Line 95 “immunogenicity or immunoreactivity occurs when the body's immune system recognizes the biomaterial as a foreign item” this is useless.

Response: Deleted as suggested.

9. Lines 105-106 “but still, switching … in terms of safety” this phrase is redundant.

Response: This shows the reason why biosimilars are not gaining popularity when compared to biologics.

10. Lines 115-130 “This article's main goal … and economic benefits”. This is too long and useless.

Response: Paragraph is shortened now.

11. The paragraphs 2.2. Risk factors and 2.3. Treatment are useless.

Response: Thanks, however, these paragraphs are important for the readers for better understanding the importance of manuscript.

12. Figures 2, 3, 4 must be limited to drugs used in oncology.

Response: We are only focusing to biosimilars used in oncology.

13. The paper must be shortened and re-organized: I suggest to discuss first the medical aspects (paragraphs 5 and 8), then the economic and regulatory aspects.

Response: We are not very sure which paragraphs the reviewer is referring here. We have already discussed medical aspects first and the economic part given under heading 3 is general information of Biosimilar not specific to oncology. Please see if I misunderstood this comment and respond appropriately.

14. Some figures and tables (i.e. after paragraph 6.6, or in paragraph 8) are not defined. Figure 9 seems to be useless.

Response: All tables and figures are now defined.

15. English language needs an extensive revision

Response: Sentences are rephrased based on your comments, wherever possible.

Reviewer 4 Report

This is a particularly useful, comprehensive, complete, and essential review of biosimilars available today and their use in Oncology. I congratulate the authors for this substantial review and for the effort they put into it. They sufficiently describe the adopted guidelines concerning their use in the developed regions of the world, and also the current therapeutic trends regarding the use of biosimilars in clinical oncology. The knowledge of safety-related parameters of the available biosimilars, the low cost compared to equivalent products, as well as the knowledge of other advantages will increase the acceptance rates of these products by clinicians around the world. The adoption of biosimilars by clinicians will increase the proportion of patients having access to these treatments, thereby saving thousands of human lives annually. I would like the issue of ways and means to increase the percentage of physicians who will trust biosimilars to be elaborated by the authors a bit more.

Author Response

This is a particularly useful, comprehensive, complete, and essential review of biosimilars available today and their use in Oncology. I congratulate the authors for this substantial review and for the effort they put into it. They sufficiently describe the adopted guidelines concerning their use in the developed regions of the world, and also the current therapeutic trends regarding the use of biosimilars in clinical oncology. The knowledge of safety-related parameters of the available biosimilars, the low cost compared to equivalent products, as well as the knowledge of other advantages will increase the acceptance rates of these products by clinicians around the world. The adoption of biosimilars by clinicians will increase the proportion of patients having access to these treatments, thereby saving thousands of human lives annually. I would like the issue of ways and means to increase the percentage of physicians who will trust biosimilars to be elaborated by the authors a bit more.

Response: Many thanks to the learned reviewer for the appreciation. As per the sagacious suggestions, the information related to the issue of ways and means to increase the percentage of physicians who will trust biosimilars has been elaborated in the future perspective section (last paragraph of future perspective section).